# Perfluorooctanoic Acid (PFOA) Exposure and Abnormal Alanine Aminotransferase: Using Clinical Consensus Cutoffs Compared to Statistical Cutoffs for Abnormal Values

**DOI:** 10.3390/toxics11050449

**Published:** 2023-05-10

**Authors:** Alan Ducatman, Youran Tan, Brian Nadeau, Kyle Steenland

**Affiliations:** 1School of Public Health, West Virginia University, Morgantown, WV 26506-9190, USA; 2Gangarosa Department of Environmental Health, Rollins School of Public Health, Emory University, Atlanta, GA 30322, USA; 3Department of Gastroenterology, William Beaumont Hospital, Royal Oak, MI 48173, USA

**Keywords:** perfluoroalkyl substances (PFAS), perfluorooctanoic acid (PFOA), chemical and drug induced liver injury, alanine amino transferase (ALT), fatty liver, nonalcoholic fatty liver disease (NAFLD), reference values, biomarkers

## Abstract

Background: Per- and polyfluoroalkyl substances (PFASs) including perfluorooctanoic acid (PFOA) are ubiquitous environmental contaminants. Prior analysis in the large “C8 Health Project” population defined abnormal alanine aminotransferase (ALT) with statistically derived cutoffs (>45 IU/L in men, >34 IU/L in women). Objective: To explore the degree to which PFOA was associated with modern, clinically predictive ALT biomarker cutoffs in obese and nonobese participants, excluding those with diagnosed liver disease. Methods: We reevaluated the relationship of serum PFOA to abnormal ALT using predictive cutoff recommendations including those of the American College of Gastroenterology (ACG). Evaluations modeled lifetime cumulative exposure and measured internal PFOA exposure. Results: ACG cutoff values (≥34 IU/L for males, ≥25 IU/L for females) classified 30% of males (3815/12,672) and 21% of females (3359/15,788) above ALT cutoff values. Odds ratios (OR) for above cutoff values were consistently associated with modeled cumulative and measured serum PFOA. Linear trends were highly significant. ORs by quintile showed near monotonic increases. Trends were stronger for the overweight and obese. However, all weight classes were affected. Conclusion: Predictive cutoffs increase the OR for abnormal ALT results. Obesity increases ORs, yet association with abnormal ALT pertains to all weight classes. The results are discussed in context of current knowledge about the health implications of PFOA hepatotoxicity.

## 1. Introduction

The aim of this work is to characterize the contribution of PFOA to ALT values above clinically recommended cutoffs, in men and women in a previously studied population [1]. Previous studies addressed older, less predictive, statistically derived (arbitrary) cutoffs. A subsidiary goal, not previously addressed in a large cohort with a similarly wide range of PFOA exposure, nor with modeled cumulative exposure, is to identify if such associations pertain to normal, overweight, and obese participants as defined by body mass index (BMI).

PFOA is a widespread and highly persistent environmental contaminant, best known for its association with the manufacture of nonstick cookware but also one of a large class of related PFASs used to make a very wide variety of useful products [2]. As a result of manufacturing, commercial use, and household product use, PFOA and other PFASs have been widely distributed into the environment from factory emissions or household waste and sewage, so that virtually every human being has detectable PFOA (and other PFASs) in serum internationally [2,3,4]. PFOA and other long chain (carbon chain length ≥ 6) perfluoroalkyls such perfluorooctane sulfonate (PFOS) are also concentrated in other human organs, notably the liver [5,6,7], and they have been associated with health outcomes that include deleterious effects on lipid metabolism, cancer outcomes such as kidney cancer, and immune outcomes such as reduced vaccine uptake [8,9]. The US Environmental Protection Agency has proposed that PFOA and PFOS be designated as hazardous substances [10]. Japan has announced the intention to ban the use of PFOA in manufacture and also the import of a wide variety of products containing PFOA [11].

Several lines of evidence support PFOA and other PFASs as a cause of disrupted liver metabolism. Experimental PFOA exposure causes higher blood transaminases including higher ALT, as well as enlarged hepatocytes with steatosis across species in experiments [8,12,13]. Humans are also affected. Cross-sectional and longitudinal studies from populations around the world have associated PFOA and other long chain perfluoroalkyls with adversely higher liver transaminases, which are markers of liver damage, as well as with higher total and LDL cholesterol [8,9,14,15]. Elevated cholesterol is relevant to the discussion of hepatotoxicity because the liver is the principle organ site for the control of cholesterol homeostasis [16]. The combination of consistently higher serum cholesterol and higher biomarkers of liver damage such as aspartate aminotransferase (AST), gamma glutamyl transferase (GGT), and especially ALT is suggests a toxicant-induced nonalcoholic fatty liver disease (NAFLD) mechanism for PFAS hepatoxicity [8,17,18,19,20,21,22,23].

Adult human evidence for an association of PFAS exposure with higher liver transaminases including ALT has been present in the human literature since 2007, and pertains to populations from North America, Europe, and the Asia-Pacific region including Japan, China, and Korea [1,21,24,25,26,27,28,29,30]. The human population evidence is mostly but not entirely consistent. Two small worker population studies by 3M researchers did not report associations of ALT with past [31] or to recent [32] PFOA exposure. In the Canadian Health Survey (*n* = 2288 in several survey periods) PFOA exposure was not associated with ALT in cross-sectional data but PFOA was associated with other biomarkers of hepatotoxicity such as 16.5% higher GGT, and a summed mixture of PFASs was also associated with higher levels of aspartate amino transferase AST and GGT [33]. However, the data are mostly consistent. A recent review of the PFAS hepatotoxicity literature detected 85 relevant rodent studies, and 23 relevant human studies including eight population studies that could be incorporated into a meta-analysis [13]. The meta-analysis detected a weighted relationship of PFOA to human ALT (z-score = 6.20, *p* < 0.001), including the three studies with a longitudinal design (z-score = 5.12, *p* < 0.001). Similar but slightly less strong relationships were detected for PFOS and perfluorononanoic acid (PFNA). The review also addressed the extensive experimental literature which was considered supportive [13].

The present study concerns the adult exposure population of the US mid-Ohio valley region of Ohio and West Virginia, eligible to participate in the “C8 Health Project” because of exposure in one of six PFOA-contaminated water districts. In a past publication (*n* = 47,092 adults), results adjusted for age, physical activity, body mass index (BMI), income and education, alcohol intake, and cigarette smoking, found a positive association of above-normal ALT (using older statistically derived cutoff values of 45 IU/L for men and 34 IU/L for women) with PFOA, with a steady increase in in the odds ratio (OR) of 1.10 (95% C.I. 1.07, 1.13) for each additional unit of measured log serum PFOA [25]. Investigators also developed modeled estimates of cumulative PFOA exposure in the serum over each subject’s lifetime, taking into account water levels, ingestion rates, and half-life for each person [1]. Using these modeled estimates in a subset of the original cross-sectional study (*n* = 32,354), investigators found a 16% increase in above-normal ALT values, again using statistically derived cutoff values as the above-mentioned cross-sectional study which used measured serum PFOA [1,25].

Historical, statistically-derived liver enzyme cutoffs are generally set at 2.5% of the population at the time of determination [34], which can be many years before a population study using the same cutoffs is performed. While such values are still in use in many laboratory reports, they are appreciably higher than clinically predictive cutoffs recommended by cognizant professional organizations interested in prevention, early detection, and mitigation of liver disease including nonalcoholic fatty liver disease (NAFLD) [35]. A normal transaminase test such as ALT does not rule out the presence of liver disease, nor does an elevated transaminase alone make a diagnosis of liver disease. However, elevated transaminases may engender consideration of further clinical evaluation [36], and it is understood that an elevation of measured transaminase levels in a population has adverse morbidity and mortality health outcome significance [35,37]. The concept of lower cutoffs for useful biomarkers stems from a desire to find disease early, and needs to be balanced against the cost and risk of additional testing [35]. All authorities recognize that the statistically based cutoffs are unsatisfactory because they are too high and greatly underestimate disease, yet there is no single accepted best cutoff for ALT because algorithmic approaches outperform single enzyme tests for the prediction of liver disease [35,36,37,38]. The predictive value of transaminases such as ALT of human liver function is augmented but not perfected in algorithmic evaluations that add personal risk factors, other biomarkers, and when indicated these noninvasive laboratory tests are further augmented by noninvasive imaging for the early detection of liver disease [35]. The gold standard for diagnosis is liver biopsy, an invasive test. NAFLD is considered a global pandemic, and it is also seriously underdiagnosed in clinical practice [36]. The underdiagnosis results in missed opportunities for prevention [35,38]. Obesity is an important risk factor for NAFLD [35]. The number of individuals estimated to have nascent NAFLD is remarkable. The adult prevalence is considered to be highest in the Middle East (30.45%, 95% C.I. 13.48–58.23), and the lowest in Africa (13.48%, 95% C.I. 5.69–28.69) [35]. Lean individuals comprised about 20% of the Japanese NAFLD population, underlining that the disease is more prevalent in but not limited to the overweight and obese [39].

A review article in the American Association for the Study of Liver Diseases (AASLD) journal Clinical Liver Disease recommended the values of 30 IU/L for males, and 19 IU/L for females as upper limits of normal ALT [40]. These values have been modeled in a Korean population (*n* = 426,013 subjects) with ten years of longitudinal follow-up and showed favorable performance characteristics [41]. The ACG clinical guidance for the evaluation of abnormal liver chemistries has suggested that an upper limit of normal cutoff for ALT for liver disease due to a multitude of causes is 33 IU/L for males and 25 IU/L for females [37]. We decided to employ these examples as a means to model different kinds of cutoffs that are based on predictive models rather than arbitrarily chosen statistical cutoffs, although other thoughtful recommended cutoff values are available in the literature [42]. All population derived cutoffs based on human health [37,40,41,42] are appreciably lower than older, statistically based laboratory normal values [43] that have been used in PFAS studies that evaluate the relationship of serum PFAS levels including PFOA to abnormal transaminase levels such as ALT. With an aim to evaluate if clinically predictive cutoffs would sharpen or blur the association of serum PFOA with the liver transaminase ALT in obese and nonobese populations, we reanalyzed the longitudinally reenrolled C8 Health population data using the two ALT cutoffs that have been designed for the early detection and evaluation of liver disease processes, and we have used that consideration as a departure point for a discussion concerning PFASs and hepatotoxicity.

## 2. Methods

### 2.1. Study Population

The C8 Health population was recruited from 6 water districts with variable degrees of PFOA water and human serum contamination, while levels of other common serum PFAS contaminants were similar to contemporaneous US population norms. An estimated 81% of residents participated [44]. (In the residential population, 1759 had also worked at the Dupont chemical plant, the source of the PFOA affecting regional drinking water). Participants filled out a survey containing demographic and health information, and most provided blood for serum PFAS testing as well as an extensive battery of clinical laboratory tests. Detailed methods for cohort recruitment and data collection have been published previously [1,44]. Analysis of liver biomarkers was conducted among 28,460 people age ≥20 from the C8 Health Study (conducted in 2005/2006), who were free of liver disease and had both liver enzymes including ALT and PFOA measured at that time. Participants were also interviewed in a protocol developed by the C8 Science Panel in 2008–2011, in order to determine whether they had previously developed diagnosed liver disease. Assessment accounted for retrospective serum PFOA estimates (developed by the C8 Science Panel), liver disease information, and other demographic information. An enrollment flowchart is shown in Figure 1. The follow-up surveys in 2008–2011 covered demographics, residential history, health-related behaviors, and lifetime personal history of various medical diagnoses. The present study was approved by the Emory University Institutional Review Board (IRB) as part of the overall approval of the work of the C8 Science Panel.

### 2.2. PFOA Exposure Estimates

Serum PFOA measurement and lifetime cumulative modeling procedure details have been described previously [46,47]. Cumulative PFOA serum concentration estimates (ng/mL) were calculated retrospectively for each community participant for each year of life beginning in 1952 or the participant’s birth year, whichever occurred later, through 2011. Estimates were based on historical regional data including the PFOA amounts emitted by the DuPont facility, wind patterns, river flow, and groundwater flow. Exposure estimates took into account each participant’s reported residential history, drinking-water source, tap-water consumption, workplace water consumption, and a PFOA absorption, distribution, metabolism, and excretion model. For study participants who had ever worked at DuPont plant, the exposure estimates were generated using an occupational exposure model based on historical serum PFOA measurements, participants’ work histories, and knowledge of plant operating processes [48]. These estimates were used to create a job–exposure matrix to estimate serum levels for workers across time in different jobs and departments. Residential exposure model was used for those with workers with higher residential exposures than occupational exposures. For workers whose occupational exposures were higher, serum estimates were decayed 18% per year after a person stopped working at the plant (based on an estimated mean PFOA half-life of 3.5 years for the population) [49], until they reached an exposure predicted by the residential exposure model. In addition to modeled cumulative PFOA serum concentration based on a retrospective PFOA estimate described above, we also used PFOA serum concentrations (ng/mL) measured during 2005–2006 from C8 Health Project (referred to as “measured serum PFOA”) to assess the effect of both modeled cumulative PFOA exposure and measured serum PFOA on liver biomarkers. Modeled serum cumulative estimates correlated well with measured serum values (r = 0.71) [45].

### 2.3. Measures of ALT and Other Factors

ALT was selected as the liver biomarker due to the fact that it is the most specific to liver injury and commonly elevated in hepatic steatosis. Other liver biomarkers such as AST can also be elevated in the setting of hepatic steatosis and hepatic inflammation. However, AST is not specific to liver injury and is also produced in muscle and increased in the setting of muscle injury. In addition, AST elevation is closely associated with alcohol-related liver disease, which is not the focus of our study. Additional biomarkers of liver damage such as alkaline phosphatase and bilirubin are more associated with cholestatic liver injury or biliary obstruction and less with hepatic steatosis. Moreover, alkaline phosphates are not specific to the liver and are produced in other organs such as bone, kidney, and intestines. Hyperbilirubinemia does not always signify liver injury and may be elevated in benign liver conditions, such as Gilbert syndrome. ALT was therefore the only biomarker considered the predecessor study of statistical cutoffs for both measured serum exposure as well as modeled lifetime PFOA exposure. ALT was measured in 2005–2006 for the C8 Health Project. Blood samples were collected from participants and centrifuged, aliquoted, and refrigerated before shipping on dry ice daily from each data collection site to the laboratory [44]. ALT was measured using a Roche/Hitachi MODULAR automated analyzer (Roche Diagnostics, Indianapolis, IN, USA) at a clinical diagnostic laboratory (LabCorp, Inc., Burlington, NC, USA).

Liver disease information was obtained at the time of the C8 Health Project in 2005–2006 and again during C8 Science Panel survey between 2008–2011 and validated by review of medical records obtained from providers. Participants with diagnosed liver disease were excluded from analysis. Participants in both the C8 Health Project and the subsequent C8 Science Panel studies were asked to consent to a medical record review. Self-reported demographic information including age, sex, body mass index (BMI), alcohol consumption, race, regular exercise, smoking status, education, household income, fasting status, history of working at DuPont plant, and insulin resistance were all obtained from C8 Health Survey. Insulin resistance was defined using the homeostasis model an assessment index (HOMA IR: the product of the basal glucose and the insulin levels divided by 2.25) [50].

### 2.4. Statistical Analysis

In logistic regression, we created dichotomized measures of ALT by specifying a threshold using the suggested cutoff values from two previously published clinical guidance documents: a normal ALT of <34 IU/L for males and <25 IU/L for females [37], as well as a normal ALT of <31 U/L in males and <20 U/L in females [35,40,41]. Dichotomized measures of ALT were analyzed in relation to modeled cumulative PFOA serum concentrations through 2005 or 2006 (depending on survey year), as well as in relation to measured PFOA serum concentration obtained in 2005 or 2006. Modeled cumulative exposure and measured serum PFOA were analyzed as a natural–log-transformed continuous variable and by quintiles. We used the same set of a priori covariates as precedent analysis measured at baseline in 2005 or 2006 in the C8 Health Project, including age, sex, body mass index (BMI), alcohol consumption, race, regular exercise, smoking status, education, household income, fasting status, history of working at DuPont plant, and insulin resistance [1].

To further investigate the effect of PFOA on liver biomarker across BMI classes, a separate linear regression model was run for each BMI category: underweight/normal weight, overweight, and obese. We consider the underweight and normal weight together, as the sample size for underweight is comprised of only 364 subjects. ALT has a positively skewed distribution in populations [1]. Natural-log-transformed liver function marker ALT was analyzed in relation to modeled cumulative PFOA serum concentrations through 2005 or 2006 (depending on survey year) and measured PFOA serum concentration in 2005 or 2006. Modeled and measured serum PFOA was analyzed as a natural–log-transformed continuous variable and by quintiles. Associations between PFOA and ALT were adjusted for the same set of covariates as above under different BMI classes [1].

## 3. Results

Lower cutoff results (Table 1): Using the AASLD literature ALT cutoffs [35,40] (≥30 IU/L for men, ≥19 IU/L for women) subsequently modeled in longitudinal data from S Korea [41], we reanalyzed the C8 Health Project data via logistic regression. For males in the fully adjusted lifetime PFOA cumulative model, those in the fifth quintile of modeled cumulative PFOA exposure, odds ratios were 1.12 (1.03, 1.33) and for females they were 1.22 (1.10, 1.36, compared to the referent (quintile 1). An increase in continuous modeled cumulative PFOA ((ng/mL)-years) was significantly associated with increased odds of abnormal PFOA for both men and women).

Use of measured 2005/2006 PFOA showed similar but stronger associations for the (≥30 IU/L for men, ≥19 IU/L for women) cutoff. The OR for abnormal ALT for top quintile of PFOA was 1.33 (95% C.I. 1.17, 1.52) for men and 1.29 (1.16,1.44) for women. There are again significant trends of increased risk for both males and females (based on the coefficient for the continuous variable).

Higher cutoff results (Table 2): Using the ALT abnormal cutoffs suggested by Kwo et al. for an ACG Guideline (≥34 IU/L for males and ≥26 IU/L for females) [37], for the fully adjusted cumulative PFOA exposure model, males in the fifth quintile of PFOA exposure OR was 1.205 (1.04, 1.40), and for females in the fifth quintile OR was 1.19 (1.05, 1.36) more likely to be above the cutoff. Results are again similar or slightly more prominent for measured values, 1.35 for the fifth to first quintile comparison in males (1.18, 1.55) and 1.20 in females (1.05, 1.37) (Table 2). Again, there are increasing trends of being above the cutoff, based on the continuous coefficient, for either cumulative or measured PFOA.

Linear regression model: Table 3 shows linear regression results where a continuous variable for ALT was regressed on log serum PFOA, either cumulative PFOA in 2005/2006 (the C8 Health Study) or measured serum PFOA in 2005/2006, by different categories of obesity in 2005/2006. Results are presented for both the continuous variable for PFOA as well as for PFOA quintile. Strong associations between ALT and PFOA was seen for all weight categories for both modeled and measured PFOA. For measured PFOA, trends were somewhat stronger in the overweight and obese categories.

## 4. Discussion

Using two cutoffs based on disease-predictive models, the results show a stronger trans-quintile PFOA association with above normal values than the values based on older cutoffs and used in previous analysis [1]. For example, the OR for having an abnormal ALT in the directly measured serum PFOA was 16% higher in the fifth compared to the first quartile using older lab-derived cutoffs [1], while this reanalysis found that it was 29.4% (female) to 33.3% higher in males using the >19 and >30 IU/L cutoffs, and it was 19.2% higher for females and 35.1% higher for males using the ≥26 and ≥34 cutoffs mentioned in the ACG literature. Modern physiologically derived predictive cutoffs increase the likelihood that PFOA is associated with abnormal ALT, whether the exposure is modeled over time as a cumulative PFOA exposure or concurrently measured as a serum level of PFOA. This finding pertains to a large population with a more than tenfold range of modeled cumulative PFOA exposure from the first to fifth quintile. The results are consistent with other cross-sectional and longitudinal studies of the association of ALT with PFOA and other PFAS exposure [13]. Recent literature from the Korean NHANES adult population (*n* = 1404, survey years 2015–2017) has shown that higher PFAS mixture concentrations including PFOA, PFOS, and PFNA are associated with higher ALT, as well as AST and GGT biomarker levels [21], but did not address abnormal cutoffs. It is reasonable to anticipate that more physiologic cutoffs for additional biomarkers for ALT as well as AST and GGT applied to other populations will also be consistently associated with still more abnormal results for PFOA (as well as with other PFASs associated with higher liver enzyme biomarkers) but perhaps the impact of reanalysis could be less apparent in populations with a narrower range of PFOA exposures, or in smaller populations. However, our data address only PFOA and ALT because the cumulative exposure model used for one part of this study was developed for PFOA only.

The association of PFOA with abnormal high ALT is seen for either measured serum or modeled cumulative PFOA and also pertains across normal, overweight, and obese participant subgroups, but is more prominent across quartiles in the overweight and obese in the measured serum data. This pattern suggests that the liver-injury risk of PFOA exposure is greater in obese populations, but not unique to those who are obese. This PFAS finding is consistent with literature about NFALD, which shows that the risk is higher in obesity but also exists in the normal-weight population [39,51], independent of PFAS consideration. In contrast, adult NHANES data from 2011–2114 found a statistically significant association of PFOA, PFNA, and perfluorohexanesulfonic acid (PFHxS) with ALT limited to the obese only using fully adjusted and weight-stratified models [52]. Our larger study from the C8 Health population (*n* = 8215 nonobese participants in the C8 Health population vs. 1801 in the NHANES population [52]), with its far wider range of PFOA exposures from background levels to very high serum concentrations, confirms that the cross-sectional association with abnormal ALT (and to ALT in general) pertains to all weight classes. It is more prominent in the overweight and obese but can be seen for all weight classes in a large enough population with a wide range of exposure.

A limitation of our study is that it is about a biomarker, not about diagnoses. The predecessor C8 Health population study [1] sought and did not find an increase in participant-reported and clinician-verified diagnoses of liver disease related to the cumulative PFOA exposure. This predecessor finding in the same population, of no association with reported and confirmed liver disease, is independent of cutoffs. The absence of an association with diagnosed disease raises the question of whether the increases in ALT that are found in association with adult PFOA (and other PFAS) exposure in this study and in other studies [1,13,21,24,25,27,28,52,53,54] are important to human health and disease. Our data address this question only indirectly, by showing that PFOA associations with abnormal high ALT are stronger when more modern, clinically recommended cutoffs are used. This question about the importance of the associations with biomarkers of liver distress can also be considered from experimental, clinical, and population perspectives.

Experimentally in vivo and in vitro, PFOA and other well-tested PFASs cause visible evidence of grossly enlarged liver, biomarkers of liver injury such as higher ALT, oxidative stress, apoptosis hepatocyte proliferation and increased liver weight, triglyceride build-up as well as lipid/lipid droplet accumulation and ballooning of hepatocytes, along with other evidence of steatosis, and metabolic evidence of disrupted hepatic bile acid metabolism and adipogenesis, and these effects are seen across animal species and in vitro [8,9,13,20,23,55,56,57,58,59,60,61,62]. There is parallel experimental evidence that PFASs induce triglyceride accumulation in stem cells [63]. Thus, the human population evidence for higher ALT and more abnormal ALT associated with PFOA is highly consistent with the extensive experimental findings [13].

From a clinical population perspective, the highly consistent liver biomarker findings await definitive demonstrations of associations with diagnosed conditions. This will require thoughtful and likely longitudinal study design because the large majority of patients with NAFLD do not know they have the condition [64]. An AASLD publication estimated that 96% of patients with NAFLD remain unaware of their condition [65]. Only a minority of early NAFLD patients will progress to end stage hepatic disease, yet it is very well established that even early stage disease is associated with substantial increases in morbidity and mortality from liver disease and from other causes [66]. The widespread lack of awareness of all but the more advanced stages of NAFLD logically constrains the capability of participant survey questionnaires to detect disease, while the mortality from multiple causes associated with NAFLD [66,67] may also degrade the capability of cross-sectional data to find diagnostic evidence of liver disease as an underlying cause of associated morbidity or mortality. Furthermore, the strong association of advanced liver disease with kidney injury [68,69] may create underestimation bias in cross-sectional studies due to the known association of advanced renal injury and any albuminuria with enhanced PFAS excretion and lower serum PFASs [70,71]. The lower ALT cutoffs recommended by professional societies are intended to be used to inform further testing and to increase awareness for preventive purposes, including the consideration of decision tree algorithms (including other noninvasive testing) for early detection of epidemic nonalcoholic fatty liver disease (NAFLD). The reason for further testing is that no single noninvasive laboratory test diagnoses NAFLD reliably. Partially as a result of the complexity of noninvasive diagnosis, as well as unwarranted clinical nihilism about our ability to intervene and prevent further disease, NAFLD is increasingly important but extensively underdiagnosed before its end stages [37,65,72]. It is possible but not proven that early liver disease was underdiagnosed in cross-sectional surveys such as the C8 Health study participant survey [1], and thoughtful designs will be needed to address underdiagnosis even in longitudinal studies. Nevertheless, adult PFAS population studies show reasonably consistent evidence of excess liver damage, and there is some evidence of excess liver mortality [73] and possibly excess liver cancer [73,74]. It is specifically the topic of clinically diagnosed human clinical liver diagnoses associated with PFOA and other PFASs that is unsettled and deserves additional, large scale studies with appropriate designs. One such study has begun which will follow 10,000 healthy adults for liver disease, with baseline values of PFOA and liver enzymes, albeit with a different, narrower exposure profile than seen in our study (Southeastern Liver Health Study, Available online: https://sciences.ncsu.edu/news/researchers-receive-17m-grant-to-explore-links-between-contaminants-and-liver-cancer/ (accessed on 3 March 2023)).

While additional evidence from formal diagnoses is awaited, there is accumulating inferential evidence of PFASs and liver disease. Using a composite score for risk of NAFLD in US NHANES (2005–2018), Lemei and colleagues showed that PFOA and “composite PFAS” is associated with NAFLD risk [75]. Imaging data are now available for PFAS association with liver findings. For the first time, a subset of 2017–2018 adult US NHANES data contained information concerning “VCTE”, or vibration-controlled transient elastography. VCTE noninvasively measures the speed of a mechanically-generated sheer wave to derive a replicable measure of liver stiffness, which in turn is a measure of hepatic fibrosis [76]. Using the NHANES 2017–2018 survey VCTE subsample, a Harvard-led team showed that PFAS exposures were associated with VCTE scores (*n* = 1135) in the NAFLD range (*n* = 446), with statistical significance present for those with obesity or high fat diets for perfluorohexane sulfonate (PFHxS), or those who drank ≥2 alcoholic drinks daily (women) or ≥3 alcoholic drinks daily (men) for both PFOA and PFHxS [77]. A study in a well-characterized group of biopsy-proven NAFLD patients (*n* = 105) showed altered bile acid and lipid pathways in association with FPAS exposure, notably in female NAFLD participants [59]. Independent of the PFAS exposure, reviews of the extensive NAFLD intervention literature show that noninvasive diet and exercise interventions have proved successful in populations with both biopsy-proven and imaging-inferred NAFLD [78]. This general, hopeful finding concerning early NAFLD reversibility with low-risk intervention has parallel support in the role of diet in protecting against murine PFAS liver damage [79]. Noninvasive algorithmic, imaging, and other designs that go beyond blood biomarkers should be further investigated in larger human populations, preferably with a wider range of PFAS exposure.

The study has limitations. The measured data are cross-sectional, while the lifetime exposure is modeled. The long serum half-life of PFOA and the similarity of findings for lifetime modeled PFOA exposure compared to measured serum PFOA mitigates that weakness to some extent. The ALT data in this and other PFOA or other PFAS studies are biomarkers, not diagnoses. However, it is already understood that the presence of higher ALT from multiple contributing causes in populations is associated with increased liver disease mortality and other causes of mortality and morbidity in multiple geographies [35,37,80,81,82]. Further, more specific biomarkers of human liver injury such as cytokeratin fragments and markers of disrupted bile acid metabolism are also seen following PFOA exposure in human populations [17,59]. While the linkage of PFOA and other PFASs to excess liver disease needs further investigation in large populations, the linkage of PFASs to higher ALT is increasingly firm [13].

## 5. Conclusions

Use of risk-based ALT cutoffs increases the OR for the association of PFOA with above the threshold ALT. The association is more prominent in the obese, yet present in all weight groups. The data add to the evidence that PFASs harm the human liver in exposure populations, and those data are reviewed from experimental and human population perspectives. The association is an indicator of worse liver health. It is reasonable to consider the kind of high level PFOA exposure noted in this population as a potential additional risk factor when considering the need to screen for early and potentially reversible evidence of liver disease.

## Figures and Tables

**Figure 1 toxics-11-00449-f001:**
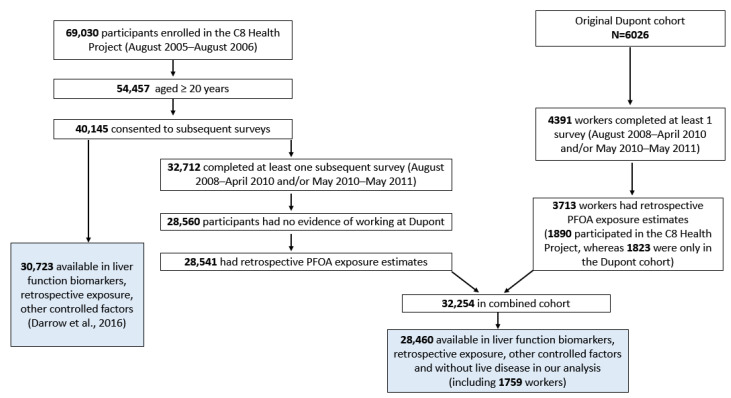
Cohort enrollment of study subjects in current analysis and Darrow’s [1] analysis, from Winquist et al. [45].

**Table 1 toxics-11-00449-t001:** ORs ^a^ for PFOA and abnormal liver enzyme ALT (cutoff >30 IU/L for males and >19 IU/L for males).

		Male (*n* = 12,672) ^d^	Female (*n* = 15,788) ^e^
**Estimated cumulative PFOA (ng/mL)**	Continuous	**1.035 (1.005, 1.066)**	**1.039 (1.015, 1.063)**
	Quintile2 ^b^	1.127 (0.999, 1.272)	1.067 (0.962, 1.183)
	Quintile3	**1.193 (1.058, 1.346)**	1.083 (0.976, 1.202)
	Quintile4	**1.216 (1.075, 1.375)**	1.044 (0.941, 1.160)
	Quintile5	**1.180 (1.025, 1.358)**	**1.219 (1.095, 1.356)**
	Test for trend ^c^	**0.01**	**0.0005**
**Measured serum PFOA in 2005–2006 (ng/mL)**	Continuous	**1.090 (1.055, 1.126)**	**1.070 (1.041, 1.099)**
	Quintile2 ^b^	**1.120 (0.992, 1.264)**	**1.145 (1.032, 1.270)**
	Quintile3	**1.194 (1.057, 1.348)**	**1.207 (1.087, 1.340)**
	Quintile4	**1.208 (1.068, 1.366)**	**1.175 (1.057, 1.306)**
	Quintile5	**1.333 (1.173, 1.516)**	**1.294 (1.162, 1.440)**
	Test for trend ^c^	**0.02**	**<0.001**

**a.** Adjusted for age, BMI, exercise, alcohol consumption, smoking status, education, insulin resistance, fasting status, history of working at DuPont plant, and race; subjects diagnosed with liver disease were excluded in the analysis. **b.** Quintile1 as reference group: Quintiles for estimated cumulative serum PFOA (in ng/mL-years): female: Q1 < 181.3; Q2 < 291.3; Q3 < 693.3; Q4 < 3236.0; Q5 ≥ 3236.0; male: Q1 < 185.14; Q2 < 303.4; Q3 < 885.1; Q4 < 4669.5; Q5 ≥ 4669.5; Quintiles for measured serum PFOA (ng/mL) in 2005–2006: female: Q1 < 9.5; Q2 < 16.6; Q3 < 30.9; Q4 < 76.6; Q5 ≥ 76.6; male: Q1 < 13.4; Q2 < 23.3; Q3 < 43.4; Q4 < 99.8; Q5 ≥ 99.8; **c.**
*p*-value for continuous PFOA variable; **d.** For male: 60% below ALT cutoff. **e.** For female: 53% below ALT cutoff.

**Table 2 toxics-11-00449-t002:** ORs ^a^ for PFOA and abnormal liver enzyme ALT (cutoff ≥ 34 IU/L for males and ≥26 IU/L for females).

		Male (*n* = 12,672) ^d^	Female (*n* = 15,788) ^e^
**Estimated cumulative PFOA (ng/mL)**	Continuous	**1.040 (1.009, 1.072)**	**1.033 (1.005, 1.062)**
	Quintile2 ^b^	1.135 (0.999, 1.290)	1.118 (0.988, 1.266)
	Quintile3	**1.244 (1.096, 1.413)**	1.056 (0.931, 1.199)
	Quintile4	**1.273 (1.118, 1.448)**	1.068 (0.940, 1.213)
	Quintile5	**1.205 (1.038, 1.398)**	**1.192 (1.048, 1.355)**
	Test for trend ^c^	**0.006**	**0.02**
**Measured serum PFOA in 2005–2006 (ng/mL)**	Continuous	**1.094 (1.057, 1.133)**	**1.043 (1.010, 1.077)**
	Quintile2 ^b^	1.128 (0.993, 1.283)	**1.150 (1.014, 1.304)**
	Quintile3	**1.233 (1.084, 1.402)**	1.116 (0.982, 1.267)
	Quintile4	**1.202 (1.054, 1.370)**	1.086 (0.955, 1.236)
	Quintile5	**1.351 (1.179, 1.549)**	**1.202 (1.056, 1.369)**
	Test for trend ^c^	**<0.001**	**0.005**

**a.** Adjusted for age, BMI, exercise, alcohol consumption, smoking status, education, insulin resistance, fasting status, history of working at DuPont plant, and race; subjects diagnosed with liver disease were excluded in the analysis. **b.** Quintile1 as reference group: Quintiles for estimated cumulative serum PFOA (in ng/mL-years): female: Q1 < 181.3; Q2 < 291.3; Q3 < 693.3; Q4 < 3236.0; Q5 ≥ 3236.0; male: Q1 < 185.14; Q2 < 303.4; Q3 < 885.1; Q4 < 4669.5; Q5 ≥ 4669.5; Quintiles for measured serum PFOA (ng/mL) in 2005–2006: female: Q1 < 9.5; Q2 < 16.6; Q3 < 30.9; Q4 < 76.6; Q5 ≥ 76.6; male: Q1 < 13.4; Q2 < 23.3; Q3 < 43.4; Q4 < 99.8; Q5 ≥ 99.8; **c.**
*p*-value for continuous PFOA variable; **d.** For male: 70.0% below ALT cutoff. **e.** For female: 79.0% below ALT cutoff.

**Table 3 toxics-11-00449-t003:** Results for linear regression of ALT on serum PFOA by category of body mass index (BMI) ^a^.

	Estimated Cumulative PFOA (ng/mL)	Measured Serum PFOA in 2005–2006 (ng/mL)
**Normal or underweight, BMI ≤ 25, *n* = 8215**		
Continuous	**0.010 (0.003, 0.016)**	**0.019 (0.012, 0.027)**
Quintile2 ^b^	**0.044 (0.014, 0.074)**	0.003 (−0.028, 0.033)
Quintile3	**0.037 (0.007, 0.068)**	0.020 (−0.010, 0.051)
Quintile4	0.030 (−0.001, 0.060)	0.029 (−0.001, 0.060)
Quintile5	**0.062 (0.031, 0.094)**	**0.068 (0.038, 0.099)**
Test for trend ^e^	**0.006**	**<0.001**
**Overweight, BMI > 25, ≤ 30, *n* = 10,074**		
Continuous	**0.011 (0.004, 0.017)**	**0.031 (0.024, 0.039)**
Quintile2 ^c^	−0.001 (−0.029, 0.027)	**0.053 (0.025, 0.081)**
Quintile3	**0.031 (0.003, 0.059)**	**0.090 (0.061, 0.118)**
Quintile4	**0.042 (0.013, 0.070)**	**0.102 (0.074, 0.131)**
Quintile5	**0.033 (0.002, 0.064)**	**0.117 (0.088, 0.147)**
Test for trend ^e^	**0.001**	**<0.001**
**Obese BMI > 30, *n* = 10,171**		
Continuous	**0.010 (0.003, 0.016)**	**0.030 (0.022, 0.038)**
Quintile2 ^d^	**0.032 (0.004, 0.060)**	**0.064 (0.036, 0.092)**
Quintile3	**0.041 (0.012, 0.069)**	**0.073 (0.044, 0.101)**
Quintile4	**0.033 (0.005, 0.062)**	**0.074 (0.046, 0.103)**
Quintile5	**0.057 (0.027, 0.088)**	**0.115 (0.086, 0.145)**
Test for trend ^e^	**0.006**	**<0.001**

**a.** Adjusted for sex, age, BMI, exercise, alcohol consumption, smoking status, education, insulin resistance, fasting status, history of working at DuPont plant, and race; subjects diagnosed with liver disease were excluded in the analysis. **b.** Among normal or underweight groups, quintile1 as reference group: for estimated cumulative serum PFOA (ng/mL): Q1 < 187.6; Q2 < 336.5.0; Q3 < 915.0; Q4 < 4531.2; Q5 ≥4531.2; for measured serum PFOA (ng/mL): Q1 < 10.5; Q2 < 19.0; Q3 < 36.2; Q4 < 86.9; ≥ 86.9; **c.** Among overweight groups, quintile1 as reference group: for estimated cumulative serum PFOA (ng/mL): Q1 < 191.3; Q2 < 315.7; Q3 < 871.9; Q4 < 4593.9; Q5 ≥ 4593.9; for measured serum PFOA (ng/mL): Q1 < 11.9; Q2 < 21.0; Q3 < 40.8; Q4 < 96.8; Q5 ≥ 96.8; **d.** Among obese groups, quintile1 as reference group: for estimated cumulative serum PFOA (ng/mL): Q1 < 174.0; Q2 < 255.7; Q3 < 583.5; Q4 < 2801.9; Q5 ≥ 2801.9; for measured serum PFOA (ng/mL): Q1 < 10.6; Q2 < 18.1; Q3 < 32.9; Q4 < 78.5; Q5 ≥ 78.5; **e.**
*p*-value for continuous PFOA variable.

## Data Availability

Data from the C8 Health project are held at four universities and are not publicly available under a legal settlement agreement. No new data were created to perform this study.

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
