# Peer review of "Perfluorooctanoic Acid (PFOA) Exposure and Abnormal Alanine Aminotransferase: Using Clinical Consensus Cutoffs Compared to Statistical Cutoffs for Abnormal Values"

_toxics, 2023, doi:10.3390/toxics11050449_

Round 1

Reviewer 1 Report

In the manuscript entitled "Perfluorooctanoic Acid (PFOA) exposure and abnormal Alanine Aminotransferase Using Clinical Consensus Guidance, Implications for Population Health", the author conducted a concise analysis of the effects of the perfluorooctanoic acid exposure on liver health. The data in this article are reliable and have certain reference value for improving the prevention awareness of such pollutants. The innovation of this study can be further improved, but the current version is also acceptable.

Minor:

The title of the manuscript seems too broad and can be further modified to more accurately summarize the content of this study.

Author Response

Reviewer:   In the manuscript entitled "Perfluorooctanoic Acid (PFOA) exposure and abnormal Alanine Aminotransferase Using Clinical Consensus Guidance, Implications for Population Health", the author conducted a concise analysis of the effects of the perfluorooctanoic acid exposure on liver health. The data in this article are reliable and have certain reference value for improving the prevention awareness of such pollutants. The innovation of this study can be further improved, but the current version is also acceptable.

Minor:

The title of the manuscript seems too broad and can be further modified to more accurately summarize the content of this study.

Author Response:   We thank the reviewer for the overall assessment.   We have more precisely described the paper in a new title, as follows:   “Perfluorooctanoic Acid (PFOA) Exposure and Abnormal Alanine Aminotransferase:  Clinical Consensus Cutoffs Compared to Statistical Cutoffs for Abnormal Values.” 

Reviewer 2 Report

The paper is well written, easy to understand, provides a quality review of the literature, and a solid depiction of its weaknesses. Overall, it has everything a manuscript needs to be published.  However, I am not convinced it has everything it needs to be a strong publication.  

The authors demonstrate that there is a correlation between PFOA concentrations and ALT concentrations. It is not an impressively strong induction, but they demonstrate this correlation and show it stays over a high number cohort and where the ALT levels might be associated with toxicity (even in obese individuals).  This is good.  However, whether ALT is the right parameter to evaluate was not assessed - as serum data of AST, LDH, cholesterol, etc. was not evaluated.  Why not these other parameters.  If not here than it should be in the Introduction. Second, Why just PFOA. PFOA is not legal for most use in the USA anymore and its levels are decreasing in human serum. Why not other PFAS, total PFAS, all legacy PFAS instead. Either these should be examined or the reasons for not doing this work provided in the Introduction.  

Author Response

The paper is well written, easy to understand, provides a quality review of the literature, and a solid depiction of its weaknesses. Overall, it has everything a manuscript needs to be published.  However, I am not convinced it has everything it needs to be a strong publication.  

The authors demonstrate that there is a correlation between PFOA concentrations and ALT concentrations. It is not an impressively strong induction, but they demonstrate this correlation and show it stays over a high number cohort and where the ALT levels might be associated with toxicity (even in obese individuals).  This is good.  However, whether ALT is the right parameter to evaluate was not assessed - as serum data of AST, LDH, cholesterol, etc. was not evaluated.  Why not these other parameters.  If not here than it should be in the Introduction. Second, Why just PFOA. PFOA is not legal for most use in the USA anymore and its levels are decreasing in human serum. Why not other PFAS, total PFAS, all legacy PFAS instead. Either these should be examined or the reasons for not doing this work provided in the Introduction.  

Author response:   We appreciate this discussion. Here is a response about both practice and theoretical reasons for the focus on ALT only, followed by the added text in response.  

Background:  The topic of whether to use GGT and AST was discussed among the authors at the inception of the paper (we did not discuss  LDH but it is the same discussion). 

The practical reason for including ALT only is simple and immutable.  The previous analysis using old-fashioned statistical cut-offs , the historic yet common cutoffs devoid of physiologic consideration,  was done only for ALT in the original analysis.  Thus, any comparison to the summed exposure model would have required a new analysis and needed additional consideration for both the old fashioned cutoffs and the modern cutoffs.   Most of the paper would be about history, and not forward looking, especially since there are already reports of the other biomarkers in other papers.  Logistically, any paper going back and remodeling both the statistical and physiologic cutoffs for three biomarkers would have been huge, and half the effort would be about history. 

(By the way, we agree strongly with the reviewer that a additional practical  efforts are needed for cholesterol/LDH cholesterol, and other lipid fractions for normal/abnormal cutoffs.   We further agree with the implicit acknowledgement that lipid fractions provide insight into liver function.   That additional work, while needed,  is of very large scope.  It needs its own paper.)   

There are also theoretical reasons we focused on ALT.   For physiologic reasons, we are unsure that a similar consideration of AST and GGT would have added much in the original paper or in this follow up.  Work with biomarkers that are less specific for non-alcoholic, noninfectious causes of steatosis and hepatitis such as AST and GGT is needed in the context of complex algorithms, but such work has already been done in several PFAS papers.  It is a related but different topic.  The theoretical reason for a new paper addressing these less specific  NAFLD biomarkers should be about the algorithms and indexes, not about the biomarkers per se.  Further, it is already known that AST and GGT are associated to PFAS exposure.  A greatly lengthened paper would simply have made the same point, two more times.    

ALT is the most specific conventional laboratory biomarker for nonalcoholic, noninfectious liver damage. (This fully defensible statement should not be confused with the related but different topic of which biomarker is most useful for hepatic severity indexes in high risk patients. We were not trying to assign risk scores)   It is not surprising that ALT is the most associated with PFAS in general and PFOA specifically.   This is shown by the work of Costello et al,  who are among a larger group of authors from multiple universities  (including two of the authors of this paper)  who have noted that GGT and AST are also associated to PFOA.   The reason to also look at AST and GGT (or LDH) is in risk indexes in individuals already regarded to have probable a priori risk  In summary,  we had both practical and theoretical reasons for limiting the analysis to our topic. 

Sophisticated readers could easily have the same question asked by the reviewer.  Here is what we added to the methods, to address the question,  without creating an excessively long digression  

“ALT was selected as the liver biomarker due to the fact that it is most specific to liver injury and commonly elevated in hepatic steatosis. Other liver biomarkers such as AST can also be elevated in the setting of hepatic inflammation and hepatic steatosis. However, AST is not specific to liver injury and is also produced in the muscle and increased in the setting of muscular injury. In addition, AST elevation is closely associated with alcohol-related liver disease, which is not the focus of our study. Additional biomarkers of liver injury such as alkaline phosphatase and bilirubin are more associated with  cholestatic liver injury or biliary obstruction and less with hepatic steatosis. Moreover, alkaline phosphatase is not specific to the liver and is produced by other organs such as bone, kidney, and intestines. Hyperbilirubinemia does not always signify liver injury and may be elevated in benign liver disease, such Gilbert's disease. Alt was the only  liver biomarker considered in the predecessor study of statistically derived cutoffs for both modeled lifetime PFOA exposure and concurrently measured exposure.  Accordingly, ALT was measured in 2005-6……”

Reviewer 3 Report

The paper is focused on studying the effects of PFOA on liver health

The work is based on data collected over several years on a large population and highlights, despite some limitations, the correlation between high levels of PFOA and liver damage

The authors use a statistical analysis of the data to highlight the need to decrease the ALT cutoff values

The work is interesting and has an important impact on preventing liver disease

Author Response

The paper is focused on studying the effects of PFOA on liver health

The work is based on data collected over several years on a large population and highlights, despite some limitations, the correlation between high levels of PFOA and liver damage

The authors use a statistical analysis of the data to highlight the need to decrease the ALT cutoff values

The work is interesting and has an important impact on preventing liver disease

Author response:  We thank the reviewer for these  comments.  We agree that the paper has a deliberately narrow data focus, and we are gratified that the reviewer appreciates the  (not at all narrow) importance of cut-offs to considerations of population health.  In addition, we think the work nicely illustrates the problem of mischaracterizing the physiology of the selected biomarker and assuming the associations are “small, ” a problem that has persisted in the literature and which is inappropriate when  one considers the impact on normal/abnormal values and the implications for population health. The association appears  meaningful in the population studied and can be seen as more meaningful when the cut-off is physiologic.
